# Establishing the content of gender stereotypes across development

**Jessica Sullivan**[1]*, **Angela Ciociolo**[2], **Corinne A. Moss-Racusin**[1]

**1** Department of Psychology, Skidmore College, Saratoga Springs, New York, United States of America,
**2** Angela Ciociolo Marketing and Design, Grafton, Massachusetts, United States of America

* jsulliv1@skidmore.edu

## Abstract

Gender stereotypes shape individuals' behaviors, expectations, and perceptions of others. However, little is known about the content of gender stereotypes about people of different ages (e.g., do gender stereotypes about 1-year-olds differ from those about older individuals?). In our pre-registered study, 4,598 adults rated either the *typicality* of characteristics (to assess *descriptive* stereotypes), or the *desirability* of characteristics (to assess *prescriptive* and *proscriptive* stereotypes) for targets who differed in gender and age. Between-subjects, we manipulated target gender (boy/man vs. girl/woman) and target age (1, 4, 7, 10, 13, 16, or 35). From this, we generated a normed list of descriptive, prescriptive, and proscriptive gender-stereotyped characteristics about people across the early developmental timespan. We make this archive, as well as our raw data, available to other researchers. We also present preliminary findings, demonstrating that some characteristics are consistently ungendered (e.g., challenges authority), others are gender-stereotypic across the early developmental timespan (e.g., males from age 1 to 35 tend to be dirty), and still others change over development (e.g., girls should be submissive, but only around age 10). Implications for gender stereotyping theory—as well as targets of gender stereotyping, across the lifespan—are discussed.

**Data Availability Statement:** Data area available on our OSF page: https://osf.io/6r4ce/?view_only=d9c59e1237e045c2be571abd94b711b6.

## Introduction

Gender plays an important role in daily life. While beliefs about gender differ within and across cultures [1]. Within particular cultures, there are often gender stereotypes (e.g., behaviors, characteristics, or attributes) that are deemed to be more normative and/or desirable for one gender than another [1, 2]. Adults in the United States who violate gender stereotypes often experience social and/or economic penalties, commonly referred to as backlash [3–12]. For example, women who violate stereotypes by self-promoting on a job interview are less likely to be hired than identical men, while men who violate stereotypes by being self-effacing were less likely to be hired than identical women [8].

While the vast majority of existing work explores backlash against adults, recent research provided the first evidence that even gender-deviant children (in this case, preschoolers) experience backlash [13]: adults report liking gender non-conforming 3-year-olds less than their

**Funding:** The authors received no specific funding for this work.

**Competing interests:** The authors have declared that no competing interests exist.

gender-conforming peers. This suggests that gender stereotypes have real-world consequences for adults and children alike. And yet, virtually nothing is known about how gender stereotypes change over development. Are stereotypic beliefs and expectations about young children the same as those about older children, adolescents, and adults? Are some traits consistently gendered across the lifespan, while others fluctuate? If they fluctuate, in what patterns? The present study investigates these questions, providing a novel assessment of how gender stereotypes change across the early developmental timespan.

Prior research demonstrates that, on average, adults in the United States believe that women should be communal (e.g., warm, supportive) and should not be dominant (e.g., aggressive, self-promoting); in contrast, men should be agentic (e.g., ambitious, independent) and should not be weak (e.g., passive, emotional) [12]. Violations of these gender stereotypes lead to backlash for adults in the U.S. (see [12] for a discussion). While the vast majority of empirical research on backlash has been conducted with participants in the United States, the limited data collected with participants in other countries including Australia [14], France [15], India [16], and China [17] reveal largely similar patterns (although more cross-cultural work is badly needed; see [18] for review).

Given the current data, it is unclear whether children in the United States, e.g., 1-year-old boys and girls, are held by adults to the same standards. Are they? If not, when in a child's development do adults begin to apply gender stereotypes? Are there previously undocumented gender-stereotypes that apply only in childhood? More broadly, do gender stereotypes remain stable as children age, or do they fluctuate across development? In the present study, we asked adults to make judgments about the typicality and desirability of a large number of characteristics for targets of a wide variety of ages. This allowed us to characterize not only the presence or absence of a particular gender stereotype but also the developmental trajectories of these stereotypes.

In general, there are three types of gender stereotypes, each of which we measured in our study. *Descriptive* stereotypes involve characteristics that are thought to be typical of a particular gender [19]. For example, women in the United States are typically viewed as more self-aware and more anxious than men, while men are typically viewed as more extroverted and forgetful than women [2]. While individuals who violate descriptive stereotypes may surprise others, these individuals generally do not encounter backlash [19]. However, the same is not true for individuals who violate prescriptive and proscriptive stereotypes. Prescriptive stereotypes describe how members of a particular gender should behave. For example, women should be communal (e.g., cheerful, patient, and interested in children), while men should be agentic (e.g., athletic, ambitious, and assertive; [2, 12]). Proscriptive stereotypes are those that describe how members of a particular gender should not behave. For example, women should not be dominant (e.g., stubborn or rebellious), while men should not be weak (e.g., emotional or yielding; [2, 12]). Of importance, people who violate prescriptive and proscriptive stereotypes typically encounter social and economic penalties (i.e. backlash; [12, 20]).

A large body of work has sought to characterize the content of descriptive, prescriptive, and proscriptive stereotypes about adult men and women. For example, Social Role Theory [21, 22] posits that stereotypes about men and women stem directly from the sex-differentiated social roles traditionally occupied by men relative to women. Due to biological predispositions in early evolutionary history, men were more likely to be hunters, and women gatherers. Over time, people ascribed role-consistent traits to men and women, and these stereotypes took on prescriptive as well as descriptive components. However, because children are not yet capable of occupying these adult roles, it is unclear how, when, and why gender stereotypes should be applied to them. Relatedly, the stereotype content model proposes that stereotypes about adults cluster around two core dimensions: competence and warmth [18, 23, 24]. People who are

viewed as high in warmth and low in competence (e.g., elderly people, housewives) elicit paternalistic stereotypes because they are perceived as low-status and non-competitive [24]. In contrast, targets who are viewed as low in warmth and high in competence (e.g., feminists, wealthy people) are met with stereotypes characterized by envy. Indeed, according to the stereotype content model, one reason that stereotype-violating women encounter backlash is that they are perceived as both high-status and competitive and therefore threaten the existing power structure. A potential challenge for this model—and one motivation for creating the current database of gender stereotypes about targets across the lifespan—is that children, by virtue of their relative lack of power, are economically, physically and socially powerless (relative to adults) and therefore are rarely thought of as threatening to the existing social order. Why, then should they experience backlash? More generally, it's not obvious that competence and warmth are the appropriate domains for characterizing children. In short, neither theory nor empirical data suggest that gender stereotypes about children and adults are necessarily identical or even similar.

In fact, the recent work that has attempted to characterize adults' gender stereotypes about children [13, 25] has found that gender stereotypes about children appear to differ—at least in some ways—from those about adults. One recent study found that gender stereotypes about appearance, toy preference, and communality may be present for toddlers but that many other types of stereotypes that apply to adults may not apply to young children [25]. For example, although adult men and elderly men are described as more intelligent than women, toddler girls and elementary-aged girls are described as more intelligent than boys [25]. This study provides an exciting and promising window into understanding how gender stereotypes differ across the lifespan; however, because it elicited ratings only for developmental categories (e.g., "adolescents (ages 12–18)"), it does not allow us to draw conclusions about developmental trajectories. Another recent study found that the gender stereotypes that apply to 3-year-old children are often meaningfully different from those that apply to adults: unlike for adults, traits that were rated as most typical for boys were rated as undesirable, and stereotypes about children were more likely to center around appearance than is typical for adults [13]. However, it is not possible to know whether some of these apparent developmental differences can be attributed to changes in gender stereotypes across the developmental timecourse, or whether they can instead be attributed to methodological differences across studies (e.g., in the stereotypes tested; in sample size; c.f. [25] which addresses some of these issues). More generally, these studies provide a promising starting point but do not provide a large dataset for future researchers to utilize and do not allow us to characterize the developmental trajectory of gender stereotypes.

Our study—which will catalogue gender stereotypes across development (e.g., ages 1–35)—is consequential for at least four reasons. First, in order for our general theories of gender stereotyping (e.g., Social Role Theory (e.g., [21, 22]) and the Stereotype Content Model [23, 24]) to be useful for understanding and predicting children's learning about gender stereotypes, they must fit the data not only for adults but also for children. Second, in order to effectively study and predict gender backlash [13, 26], it is critical that we first understand the stereotypes that underlie backlash. Third, most theoretical approaches assume that gender stereotypes are learned; this implies that stereotypes could and should change over development, although there is very little data to speak to this. Fourth, from a practical perspective, adults interact with others (including children) throughout the early developmental timespan; parents, educators, and policy-makers would do well to understand the nature of the gender stereotypes that might be guiding their interactions. While evidence does suggest that people appear to encounter backlash for violating gender stereotypes across adulthood and childhood alike (e.g., [13]), for most ages research has done little to identify what these childhood gender

stereotypes—and thus, their violations—even *are*. While we know that backlash exists, the nature of childhood gender stereotypes remain unclear, so we cannot yet predict the circumstances under which it is likely to occur across the lifespan.

In the present study, we measured adults' gender stereotypes about infants (age one), children (ages 4, 7 and 10), adolescents (age 16), and adults (age 35). To do this, we measured gendered stereotypes about targets that ranged in age from 1 to 35. We present a database of normed gender stereotypes along with pre-registered findings generated by this study. This database fills a critical gap in the literature and will provide a set of developmental norms for researchers interested in gender development, gender backlash, and gender stereotypes.

## Method

All materials, methods, and analyses were approved via Skidmore College's IRB, and pre-registered (https://osf.io/7ahks).

### Participants

Our target sample size pre-exclusions was 4,900, which we requested via TurkPrime [27]; this number was selected in order to ensure that we had approximately 100 participants per cell of our design. Participants were native English speakers who were aged 18+, who had at least a 97% approval rating for prior Mechanical Turk HITs, and who had between 100–10,000 HITs. In total, 5,260 participants consented. We did not collect demographic data from our participants and therefore cannot assess the extent to which they are representative of the general population of the United States. Consistent with our pre-registration, we excluded participants who failed to complete at least 80% of the questions ($n = 308$), and who failed any attention check ($n = 354$). This resulted in a final $N$ of 4,598.

**Design.** The current study utilized a 2 [target gender: male, female] x 2 [rating type: pre/proscriptive, descriptive] x 7 [target age: 1, 4, 7, 10, 13, 16, 35] between-subjects design. This is in contrast to previous work that has manipulated these factors within-subjects [25]. In addition, our use of particular ages (e.g., "seven-year-old") contrasts with that in previous work, which has elicited clusters of ratings (e.g., "elementary-school boys"; [25]).

We randomly assigned participants to conditions and to items within conditions. There were 175 characteristics and five attention checks, and participants were randomly assigned to see approximately 60% of the items. For each condition (e.g., for ratings of the desirability of characteristics for one-year-old boys), we obtained an average of 160 usable participants (min = 137, max = 187). For each characteristic (e.g., "pretty"), we obtained an average of 102 ratings per cell.

### Materials and procedure

Our bank of characteristics consisted of 175 unique items from previous work [2, 12, 13, 28]. These included behaviors (e.g., wrestles), traits (e.g., pretty), preferences (e.g., loves pink), and appearance-related items (e.g., wears tutus).

Participants first viewed instructions as follows: "Today you will be answering questions about how [common or typical / desirable] you think particular traits are among [age] [boys/girls/men/women]". For example, participants in one condition saw: "Today you will be answering questions about how [common or typical/desirable] you think particular traits are among 1-year-old boys".

Participants then rated each characteristic on a 1–9 Likert-Style scale with 1 indicating "not at all X" (where X was either "desirable" or "common/typical") and 9 indicating "very X"; 5 was labeled as "Neutral." They were reminded of the instructions: "Indicate how typical/

desirable it is in American society for [age, gender] to possess each of the following characteristics. [Scale = 1–9; 1 = Not at all Typical/Desirable; 9 = Very Typical/Desirable]". Characteristics were randomly ordered and randomly sampled from the body of possible characteristics described above. After rating each characteristic, we collected participants' ages and genders.

## Results

As pre-registered (and as noted above), we excluded participants who were unlikely to be attending to the task by not answering enough questions or by incorrectly completing comprehension checks. We also excluded from analyses two items that were mistakenly included in our battery: "doesn't wait his turn" (since it accidentally included a gendered pronoun) and "is clean" (since we also had the item "clean").

Two primary goals of the current work were to develop a database of gender stereotypes across the developmental early timespan and to provide these data to other researchers. To this end, our data are available on our OSF page (https://osf.io/9pgkd/).

We had three additional specific goals, each of which was pre-registered: (1) to identify items that were and weren't consistently gendered across the early developmental timespan (e.g., to find a list of gender stereotypes that characterize girls and women throughout development); (2) to identify items for which stereotypes changed over the early developmental timespan (e.g., items that are stereotypical at young ages but not at older ages); and (3) to identify items for which there were stereotypes at particular ages (e.g., to find a list of all gender stereotypes for 1-year-olds).

As pre-registered, we constructed linear models that predicted ratings for each characteristic from target gender (boy/girl), age (continuous), and their interaction. This allowed us to identify items where gender interacted with age (indicating that the presence and/or nature of the gender stereotype changed over development; these data are discussed later) and also items that were consistently gendered (i.e. there was no interaction with age, but rather a simple effect of gender).

### Items that were not gender-stereotyped

We first report items for which we found no effect of target gender and no interaction of gender and age. In other words, these were the items for which—when considering the entirety of our dataset—we had no evidence, at any age, of gender stereotyping. For ratings of typicality, 29/175 (16.6%) characteristics that showed no effect of gender are depicted in Table 1. For ratings of desirability, 59/175 characteristics (33.71%) showed no effect of gender as depicted in

**Table 1. Characteristics showing no effect of gender in ratings of typicality.**

| Characteristics | | |
|---|---|---|
| Acts as a leader | Demanding | Materialistic |
| Ambitious | Determined | Rational |
| Analytical | Extroverted | Self-centered |
| Argues with parents | Hard-working | Self-sufficient |
| Assertive | Has a strong personality | Stubborn |
| Bratty | Has business sense | Typical |
| Brave | Independent | Uncertain |
| Childlike | Is a leader | Weak |
| Decisive | Is frequently sick | Willing to take a stand |
| Defends own beliefs | Loyal | |

**Table 2. Characteristics showing no effect of gender in ratings of desirability.**

| Characteristics | | |
|---|---|---|
| Anxious | Good listener | Rational |
| Argues with parents | Has good manners | Refuses to pick up toys |
| Arrogant | Helpful | Ruthless |
| Bossy | Helps out around the house[a] | Satisfied with life |
| Bratty | Humble | Self-centered |
| Challenges authority[a] | Interrupts others | Self-critical |
| Choosy | Is frequently sick | Slobbers |
| Cold towards others | Lazy | Spiritual |
| Competent | Likeable | Stingy |
| Complicated | Loyal | Stubborn |
| Controlling | Materialistic[a] | Supportive |
| Cooperative | Moody | Thinks it's funny when other kids are crying |
| Cynical | Nosy | |
| Disobedient | Obedient | Uncertain |
| Does not use harsh language | Open minded | Unemotional |
| Enthusiastic | Persuasive | Waits turn |
| Excitable | Picky eater | Was an easy baby |
| Frequently has a runny nose | Polite | Well-behaved |
| Friendly | Prejudiced | Wholesome |
| Gets pushed around by other kids | Pulls other kids' hair | Yielding |

Note

[a] Despite the lack of overall effects of gender, characteristic showed a pairwise difference at least one age.

Table 2. In other words, there is no evidence that these 29 traits constitute descriptive gender stereotypes (Table 1), or that these 59 traits constitute prescriptive or proscriptive gender stereotypes (Table 2).

## Stereotypes that were consistent across development

**Ratings of typicality (i.e., Descriptive stereotypes).** We first considered items for which we found a simple effect of target gender (For brevity, we do not report main effects of age in the main paper, although readers may access our data in our repository (https://osf.io/9pgkd/); items described here with a main effect of gender may also have shown main effects of age; only those items for which (a) there was an age*gender interaction (discussed later) or (b) no main effect of gender are excluded from the reporting below.). Our preliminary analysis revealed 93 items for which there was a simple effect of target gender (and no interaction with age) on ratings of typicality. In other words, ratings of typicality for these items differed depending on whether the target was a boy or a girl, and this gender difference did not depend on target age. As pre-registered, we identified the items for which the Cohen's $d$ effect size of the comparison of ratings for boys vs. girls was larger than 0.4; this is in keeping with past work [2, 13]. This yielded 25 stereotypes about typicality that persisted across the developmental timeline; 15 of these met our pre-registered criteria for being descriptive stereotypes (effect size larger than 0.4 *and* a mean typicality above 6), while 10 did not (these met our effect size criteria but not the mean rating criteria and therefore were simply relatively more common for one gender than the other; we thus refer to them as "more typical" rather than "descriptive"). These are displayed in Tables 3 and 4.

**Table 3. Characteristics rated as consistently more typical in boys/men than girls/women across the lifespan.**

| Characteristic | *M* (male) | *M* (female) | Classification for boys/men | Cohen's *d* |
|---|---|---|---|---|
| Rowdy | 6.54 | 5.11 | Descriptive | 0.74 |
| Willing to take risks | 6.65 | 5.79 | Descriptive | 0.46 |
| Competitive | 6.60 | 5.87 | Descriptive | 0.40 |
| Handsome[a] | 5.79 | 3.82 | More typical | 0.97 |
| Dirty[b] | 5.74 | 3.88 | More typical | 0.90 |
| Aggressive | 5.43 | 4.30 | More typical | 0.58 |
| Sometimes hits others | 5.52 | 4.50 | More typical | 0.48 |
| Has bruised knees | 5.64 | 4.72 | More typical | 0.41 |

*Note.* All traits included here are ones for which the Cohen's *d* effect size comparing ratings for boys/men vs. girls/women was larger than 0.4. Traits classified as descriptive demonstrated a mean typicality rating for boys/men above 6, while those classified as more typical demonstrated a mean below 6.

[a]Characteristic is also a prescription for boys (see Table 5).

[b]Characteristic is also a proscription for girls (see Table 5).

## Ratings of desirability (i.e., Prescriptive and proscriptive stereotypes)

Our preliminary analysis also revealed 85 items that showed a simple effect of gender for ratings of desirability. In other words, these items were rated as consistently more desirable for one gender than the other, and this gender difference did not depend on target age. Of these, 12 met our threshold for effect size (Table 5). We found 4 items that were prescriptive for boys/men and 2 for girls/women; these were the characteristics for which there was an effect size of at least 0.4 and a desirability rating above 6. We also found 2 items that were more

**Table 4. Characteristics rated as consistently more typical for girls/women than boys/men across the lifespan.**

| Characteristic | *M* (male) | *M* (female) | Classification for girls/women | Cohen's *d* |
|---|---|---|---|---|
| Enjoys wearing skirts and dresses | 2.23 | 6.35 | Descriptive | 2.28 |
| Loves pink | 2.70 | 6.18 | Descriptive | 1.89 |
| Pretty | 3.86 | 6.59 | Descriptive | 1.43 |
| Gentle | 4.74 | 6.07 | Descriptive | 0.76 |
| Tender[a] | 4.86 | 6.05 | Descriptive | 0.66 |
| Affectionate | 5.52 | 6.51 | Descriptive | 0.55 |
| Pays attention to appearances | 4.82 | 6.14 | Descriptive | 0.55 |
| Loves children | 5.09 | 6.10 | Descriptive | 0.53 |
| Sweet | 5.51 | 6.41 | Descriptive | 0.52 |
| Warm | 5.48 | 6.34 | Descriptive | 0.52 |
| Caring | 5.53 | 6.33 | Descriptive | 0.48 |
| Flatterable | 5.41 | 6.21 | Descriptive | 0.40 |
| Graceful[a] | 3.59 | 5.22 | More typical | 0.88 |
| Clean | 4.33 | 5.82 | More typical | 0.76 |
| Helps mom bake | 3.91 | 5.39 | More typical | 0.66 |
| Fragile[b] | 4.54 | 5.52 | More typical | 0.46 |
| Enjoys cooking | 3.86 | 4.84 | More typical | 0.46 |

*Note.* All traits included here are ones for which the Cohen's *d* effect size comparing ratings for girls/women vs. boys/men was larger than 0.4. Traits classified as descriptive demonstrated a mean typicality rating for girls/women above 6, while those classified as more typical demonstrated a mean below 6.

[a]Characteristic is also a prescription for girls (see Table 5).

[b]Characteristic is also a proscription for boys (see Table 5).

**Table 5. Characteristics that were consistently rated as more desirable for one gender than the other.**

| Characteristic | M (male) | M (female) | Classification | Gender | Cohen's d |
|---|---|---|---|---|---|
| Handsome[a] | 6.92 | 4.59 | Prescription | Boys/men | 1.12 |
| Likes to play with tools | 6.50 | 4.97 | Prescription | Boys/men | 0.82 |
| Loves sports | 6.72 | 5.56 | Prescription | Boys/men | 0.67 |
| Athletic | 6.96 | 6.11 | Prescription | Boys/men | 0.48 |
| Has a big appetite | 5.91 | 4.76 | More desirable | Boys/men | 0.64 |
| Loves to get dirty | 5.76 | 4.63 | More desirable | Boys/men | 0.54 |
| Fragile[b] | 2.82 | 3.84 | Proscription | Boys/men | 0.52 |
| Graceful[b] | 5.49 | 6.90 | Prescription | Girls/women | 0.77 |
| Tender[b] | 5.85 | 6.68 | Prescription | Girls/women | 0.44 |
| Soft spoken | 4.78 | 5.55 | More desirable | Girls/women | 0.41 |
| Dirty[a] | 3.29 | 2.39 | Proscription | Girls/women | 0.45 |
| Has unbrushed hair | 4.02 | 3.22 | Proscription | Girls/women | 0.40 |

*Note*. All traits included here are ones for which the Cohen's *d* effect size comparing ratings for girls/women vs. boys/men was larger than 0.4. Traits classified as prescriptive demonstrated a mean desirability rating above 6, while those classified as proscriptive demonstrated a mean below 4. Items that met our effect size criteria but that displayed means above 4 and below 6 are described as "more desirable" for a particular gender.
[a]Characteristics that were also rated as descriptive/more typical for boys (see Table 3).
[b]Characteristics that were also rated as descriptive/more typical for girls (see Table 4).

desirable for boys/men and 1 for girls/women (*n* = 1); these characteristics met our threshold for effect size but were neither rated as especially desirable (rating above 6) or undesirable (rating below 4); we refer to these as "more desirable" rather than "prescriptive." Finally, we found traits that were proscriptive for girls/women (*n* = 2), and traits that were proscriptive for boys/men (*n* = 1); these characteristics met our threshold for effect size and had a mean desirability rating below 4.

In our previous work, we demonstrated that characteristics descriptive of 3-year-old boys also tended to be rated as undesirable (i.e., below the midpoint of our 9-point desirability scale), while the opposite was true for 3-year-olds girls [13]. We extend this finding in the present dataset; the mean desirability rating for the desirable characteristics in Table 5 for boys was 4.29 (i.e., undesirable), while it was 6.08 (i.e. desirable) for girls; these values differed significantly (*p* = .003). These data suggest that the characteristics that describe boys/men are rated as less desirable than those that describe girls/women (and, in fact, are rated as undesirable) across the lifespan. Further, we highlight that there were noticeably fewer traits viewed as consistently typical for boys/men (8) than for girls/women (17).

## Stereotypes that change over development

We next consider the characteristics for which we found a significant gender by age interaction. These were the items for which the magnitude of the gender gap changed across the developmental timeline—in other words, traits that are gender stereotypic, but as a function of target age. We found 43 characteristics where gender differences in ratings of typicality interacted with age (Table 6) and 22 characteristics where gender differences in ratings of desirability interacted with age (Table 7).

Our next goal was to understand the nature of these interactions. To do this, as pre-registered, we visualized each interaction. We then exploratorily qualitatively clustered characteristics based on shared developmental patterns. To do this, the lead author clustered the visualizations (see Fig 1) based on visual similarity, and the other two authors checked the

**Table 6. Characteristics for which gender differences in ratings of typicality interacted with age.**

| Cluster classification and included characteristics | | Gender more typical |
|---|---|---|
| Childhood gender differences | | |
| | Likes superheroes* | Boys/men |
| | Pretend to be a soldier* | Boys/men |
| | Likes princesses* | Girls/women |
| | Likes to play with dolls* | Girls/women |
| | Wears Tutus* | Girls/women |
| Adolescent gender differences—boost | | |
| | Has a big appetite | Boys/men |
| | Has unbrushed hair | Boys/men |
| | Smelly | Boys/men |
| | Emotional* | Girls/women |
| | Anxious | Girls/women |
| | Cries often | Girls/women |
| | Melodramatic | Girls/women |
| | Moody | Girls/women |
| | Pays attention to what other people are wearing | Girls/women |
| | Self-critical | Girls/women |
| Adolescent gender differences—reduction | | |
| | Likes to play outside | Girls/women |
| | Wears clothes that don't match | Girls/women |
| | Likes to be held* | Girls/women |
| | Choosy | Girls/women |
| | Snuggly | Girls/women |
| Fluctuating gender differences | | |
| | Comforts other children when they are crying | Girls/women |
| | Compassionate | Girls/women |
| | Eager to soothe hurt feelings | Girls/women |
| | Good listener | Girls/women |
| | Helps out around the house | Girls/women |
| | Sensitive to the needs of others | Girls/women |
| | Sympathetic | Girls/women |
| Cluster classification and included characteristics | | Gender more typical |
| Fluctuating gender differences | | |
| | Understanding | Girls/women |
| Direction switches | | |
| | Is submissive* | Direction switches |
| | Spiritual | Direction switches |
| | Stingy | Direction switches |
| Differences emerge late | | |
| | Childlike | Boys/men |
| | Strong | Boys/men |
| Persistent gender differences | | |
| | Masculine* | Boys/men |
| | Likes to play with tools | Boys/men |
| | Loves to get dirty | Boys/men |
| | Feminine* | Girls/women |
| | Likes to wear nail polish* | Girls/women |

(*Continued*)

**Table 6.** (Continued)

| Unclassified | | |
|---|---|---|
| | Sensitive* | N/A |
| | Steals toys* | Boys/men |
| | Intimidating | Boys/men |
| | Is easily frightened | Girls/women |
| | Messy | N/A |
| | Strong | N/A |
| | Unemotional | Boys/men |

*Note.* Asterisks indicate that characteristic also had a pre/proscriptive interaction (see Table 7).

clustering. Due to the subjective and qualitative nature of this classification process, the resulting clusters should be interpreted as useful ways of digesting our otherwise exceptionally dense dataset, and as helpful jumping-off points for future research. Fig 1 defines and depicts each cluster. Every characteristic and its cluster are depicted in Tables 6 (for ratings of typicality) and 7 (ratings of desirability).

**Table 7. Characteristics for which gender differences in ratings of desirability interacted with age.**

| Classification and included characteristics | | Gender more desirable |
|---|---|---|
| Childhood gender differences | | |
| | Plays with trucks | Boys/men |
| | Likes superheroes* | Boys/men |
| | Pretend to be a soldier* | Boys/men |
| | Steals toys* | Boys/men |
| | Likes to play with dolls* | Girls/women |
| | Sensitive* | Girls/women |
| Differences emerge late | | |
| | Ambitious | Boys/men |
| | Adorable | Girls/women |
| | Flatterable | Girls/women |
| | Is submissive* | Girls/women |
| | Likes to be held* | Girls/women |
| Persistent gender effects | | |
| | Masculine* | Boys/men |
| | Enjoys wearing skirts and dresses | Girls/women |
| | Loves pink | Girls/women |
| | Pretty | Girls/women |
| | Feminine* | Girls/women |
| | Likes princesses* | Girls/women |
| | Likes to wear nail polish* | Girls/women |
| | Wears Tutus* | Girls/women |
| Unclassified | | |
| | Demanding | N/A |
| | Dominant | N/A |
| | Emotional* | N/A |

*Note.* Asterisks indicate that characteristic also had a descriptive interaction (see Table 6).

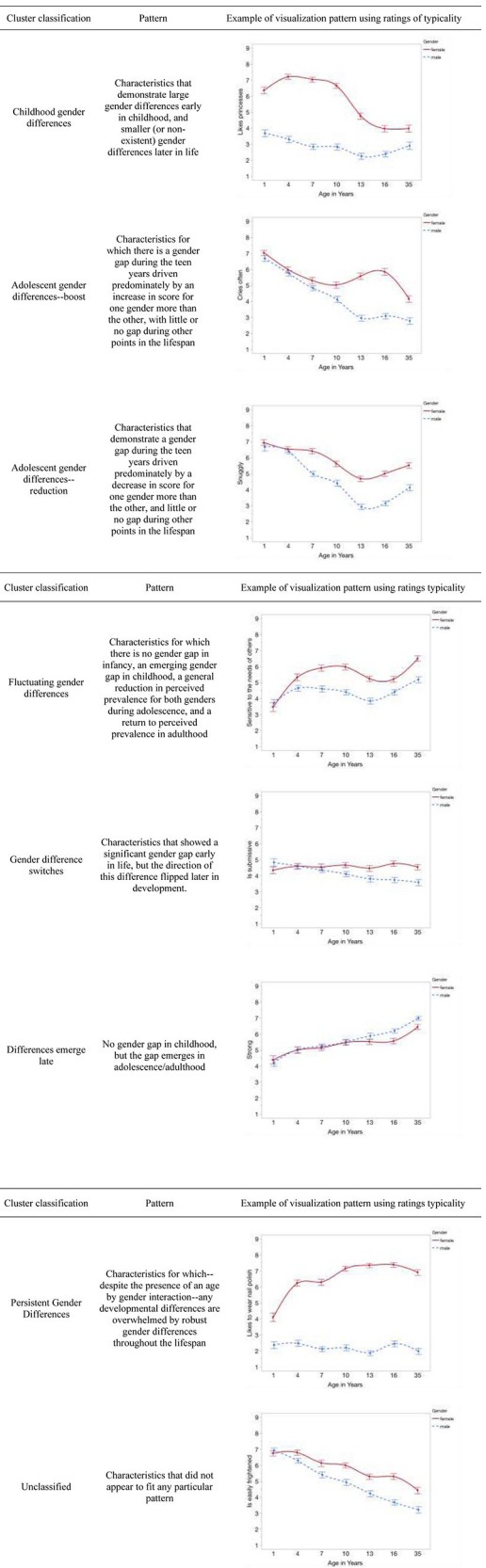

**Fig 1. Observed clusters and their patterns.**

## Stereotypes at each age

As pre-registered, for each age, we classified each item according to whether it met the criteria for being a descriptive (mean rating of at least 6 for one gender, effect size of at least .4 for the difference in ratings for boys/men vs. girls/women), prescriptive (mean rating of at least 6 for one gender, effect size of at least .4 for the difference in ratings for boys/men vs. girls/women), or proscriptive stereotype (mean rating of less than 4 for one gender, effect size of at least .4 for the difference in ratings for boys/men vs. girls/women). We did this for every item whether or not there was a significant interaction of age and gender in the analyses above; this was because we decided, *a priori*, that it was important to identify individual stereotypes at each age. Of course, as with all situations with such a large number of comparisons, even though our threshold was not statistical significance, we encourage readers to be cautious in interpreting each particular effect as it is likely that some of these effects emerged by chance alone.

This process of classification had three main outcomes. First, we assessed how many characteristics were classified as gender stereotypes at each age (see Fig 2). Interestingly, stereotypes were most frequent between the ages of 7 and 16, peaking at age 10. While the rate of prescriptive stereotypes appeared approximately the same across the developmental timespan, proscriptive and descriptive stereotypes were most frequent during childhood in our dataset. In other words, for female targets, there were more stereotypes applied to 7-year-olds than to adults, to 10-year-olds than to adults, to 13-year-olds than to adults, and to 16-year-olds than to adults. This is particularly striking because much of the existing research has focused on understanding gender stereotypes only about adults.

Next, we explored whether there were more de-, pre-, or pro-scriptions for boys/men relative to girls/women and whether the frequencies of these stereotypes changed over the early

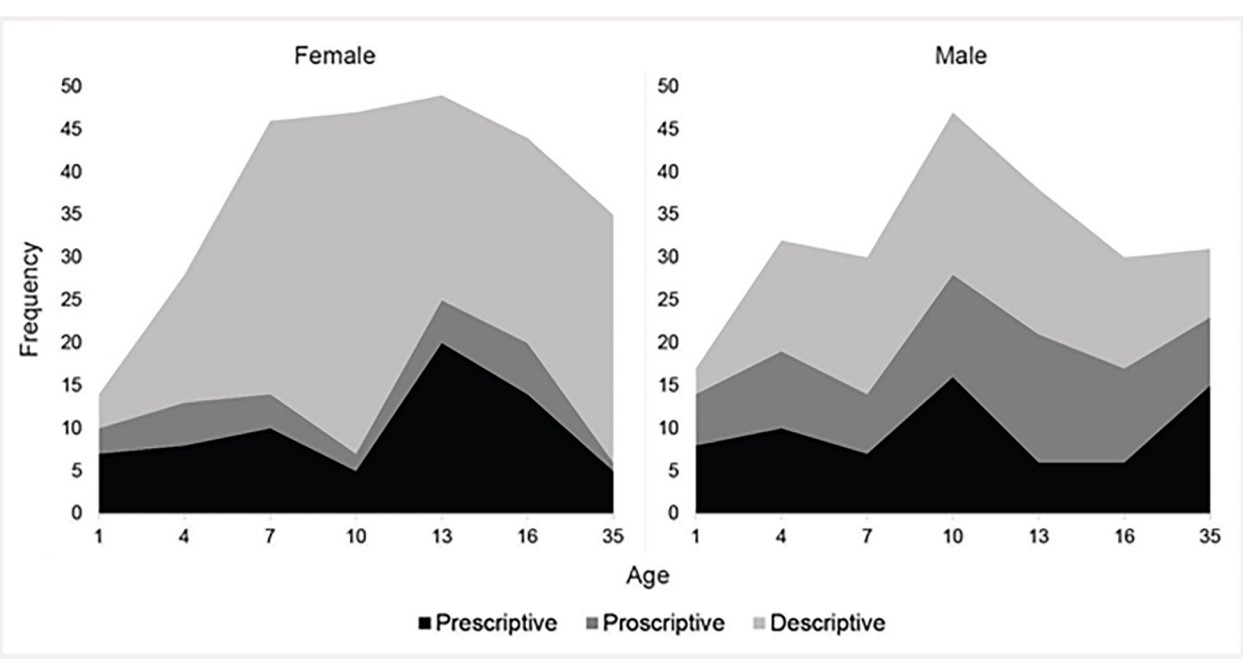

**Fig 2. Number of characteristics classified as gender stereotypes at each age by type of stereotype.** *Notes.* Y-axis is a count of stereotypes that met our pre-registered stereotype threshold. Black indicates prescriptive stereotypes, dark gray indicates proscriptions, and light gray indicates descriptive stereotypes. Note that some characteristics are double-counted (e.g., if an item was a descriptive and prescriptive stereotype at a particular age, it contributes to both the descriptive count and the prescriptive count). These data suggest that gender stereotypes are prevalent across the developmental timeline, and that children—not adults—may be subject to the most gender stereotypes.

**Table 8. Characteristics that were rated as descriptive of one-year-old children.**

| Characteristic | $M$ (girl) | $M$ (boy) | $d$ | Gender | Other Ages | | | | | |
|---|---|---|---|---|---|---|---|---|---|---|
| | | | | | 4 | 7 | 10 | 13 | 16 | 35 |
| Plays with trucks[a] | 4.43 | 6.77 | 1.13 | Boys | x | x | x | | | |
| Handsome[a] | 4.00 | 6.00 | 0.83 | Boys | | | | | | x |
| Dirty[d] | 5.28 | 6.21 | 0.42 | Boys | x | | x | x | | |
| Likes princesses[b,c] | 6.36 | 3.63 | 1.32 | Girls | x | x | x | | | |
| Pretty[b] | 6.84 | 4.98 | 0.90 | Girls | x | x | x | x | x | x |
| Likes to play with dolls[b] | 6.28 | 4.54 | 0.86 | Girls | x | x | | | | |
| Does not use harsh language | 7.65 | 6.42 | 0.48 | Girls | | x | x | | | |

*Note.* "Other ages" column indicates the other target ages (4, 7, 10, 13, 16, 35) for which this characteristic was also descriptive.

[a]Characteristic was also a prescription for boys (see Table 9).

[b]Characteristic was also a prescription for girls (see Table 9).

[c]Characteristic was also a proscription for boys (see Table 9).

[d]Characteristic was also a proscription for girls (see Table 9).

developmental timespan (Table 9). We found that a larger proportion of descriptive stereotypes were about girls/women (65.4%) than about boys/men (34.6%; $p < .0001$). A larger proportion of proscriptive stereotypes were about boys/men (72.3%; $p < .0001$) than about girls/women (27.7%); there were no effects of age on the distribution of stereotypes in either of these cases (all $p > .10$). Interestingly, this gendered asymmetry did not emerge for prescriptive stereotypes, which were equally frequent for boys/men (49.6%) and girls/women (50.4%, $p = .93$). Again, there were no effects of age on the gendered distribution of these stereotypes. These data suggest that while girls/women may consistently be subject to a relatively higher proportion of descriptive stereotypes, boys/men are subject to a higher proportion of proscriptive stereotypes throughout the lifespan.

The final outcome is a table of all items that meet the criteria for being de-, pre-, or proscriptive at each age range. These are available in our repository. To illustrate these findings for one age group, in Tables 8 and 9, we pull out all descriptive (Table 8) and prescriptive (Table 9) stereotypes for one-year-olds.

## Discussion

In the present study, we measured adults' stereotypes about male and female targets across the early developmental timespan (i.e. from infancy through early adulthood). To do this, we presented over 4,000 adults with a list of characteristics and asked them to rate either the desirability or typicality of those characteristics. Critically, participants rated the characteristics for targets that were either male or female and that were either 1, 4, 7, 10, 13, 16, or 35-years-old. This allowed us to develop the largest known normed database of gender stereotypes and to shed light on several questions about how descriptive, prescriptive, and proscriptive gender stereotypes change across the developmental timeline.

Rather than demonstrating stable stereotypic expectations for boys, girls, men, and women throughout the lifespan, our data revealed numerous developmental trends in the nature of gender stereotypes. First, items that were consistently gendered (main effects) were very rare; less than 10% of our items were consistently rated as being descriptive or pre/pro-scriptive stereotypes. Further, 29 characteristics were never descriptive of either gender, and 59 were never pre/pro-scriptive of either gender. These data suggest that theories of gender stereotypes need to take into account the fact that stereotypes are applied differently to targets of different ages

**Table 9. Characteristics that were rated as prescriptive/proscriptive of one-year-old children.**

| Characteristic | M (girl) | M (boy) | Effect size | Classification | Other ages where effect was found | | | | | |
|---|---|---|---|---|---|---|---|---|---|---|
| | | | | | 4 | 7 | 10 | 13 | 16 | 35 |
| Masculine | 3.06 | 6.01 | 1.48 | Prescriptive of Boys; Proscriptive of Girls | x | x | x | x | x | x |
| Handsome* | 4.36 | 6.79 | 1.19 | Prescriptive of Boys | x | x | x | x | x | x |
| Plays with trucks* | 4.77 | 6.51 | 0.96 | Prescriptive of Boys | x | x | x | | | |
| Likes to play with tools | 4.98 | 6.54 | 0.80 | Prescriptive of Boys | x | x | x | x | x | |
| Athletic | 4.90 | 6.22 | 0.61 | Prescriptive of Boys | x | x | | x | | |
| Loves sports | 5.11 | 6.16 | 0.56 | Prescriptive of Boys | x | x | x | x | x | x |
| Likes superheroes | 5.36 | 6.40 | 0.53 | Prescriptive of Boys | x | x | x | | | |
| Strong personality | 5.66 | 6.51 | 0.44 | Prescriptive of Boys | | | | | | |
| Enjoys wearing skirts and dresses | 6.27 | 2.82 | 1.71 | Prescriptive of Girls; Proscriptive of Boys | x | x | x | x | x | x |
| Feminine | 6.28 | 2.99 | 1.71 | Prescriptive of Girls; Proscriptive of Boys | x | x | x | x | x | x |
| Likes princesses* | 6.21 | 3.80 | 1.20 | Prescriptive of Girls; Proscriptive of Boys | x | x | x | x | x | x |
| Likes to play with dolls* | 6.58 | 4.06 | 1.25 | Prescriptive of Girls | x | x | x | x | x | |
| Pretty* | 6.83 | 4.61 | 1.08 | Prescriptive of Girls | x | x | x | x | x | x |
| Graceful | 6.76 | 5.51 | 0.69 | Prescriptive of Girls | x | x | x | x | x | x |
| Sensitive | 6.17 | 5.31 | 0.43 | Prescriptive of Girls | | | x | | | |
| Characteristic | M (girl) | M (boy) | Effect size | Classification | Other ages where effect was found | | | | | |
| | | | | | 4 | 7 | 10 | 13 | 16 | 35 |
| Likes to wear nail polish | 4.84 | 2.85 | 0.99 | Proscriptive of Boys | x | x | x | x | x | x |
| Wears Tutus | 5.30 | 3.34 | 0.96 | Proscriptive of Boys | x | x | x | x | x | |
| Loves pink | 5.66 | 3.87 | 0.86 | Proscriptive of Boys | x | x | | x | x | |
| Dirty* | 2.56 | 3.54 | 0.47 | Proscriptive of Girls | x | x | | x | x | |
| Challenges authority | 3.12 | 4.07 | 0.41 | Proscriptive of Girls | | | | | | |

*Note.* "Other ages" column indicates the other target ages (4, 7, 10, 13, 16, 35) for which this characteristic was also descriptive.

*Characteristic was also a description for that age and gender (see Table 8).

—an idea that has not received significant attention in the literature thus far (to our knowledge).

The existence of a sizable subset of ungendered characteristics suggests that demand characteristics were unlikely to be responsible for our findings. In interpreting these results, it is important to note that we selected each of our 175 target characteristics from the existing literature [2, 12, 13, 28]—these were items for which we had strong reason to believe that stereotypes might emerge. Indeed, for some items that are considered relatively central to defining particular gender stereotypes, we found no effects whatsoever of gender (e.g., there were no gendered effects on ratings of desirability for items like *helping*, *wholesome*, *is a leader*, *bossy*, *challenges authority*, *controlling*, *moody*, *friendly*, *good listener*, *competent* or *polite*, all of which are items that previous work has suggested may be gendered). These data highlight the importance of empirically testing the presence or absence of gendered stereotypes one-at-a-time (in contrast to some previous work, which has clustered traits; see [25]).

While there was a sizeable subset of traits for which there was no evidence of gender stereotyping, the majority of traits did show some evidence of gendering (it is important to note that many of these effects, while statistically significant, did not reach our pre-registered effect size criteria and therefore are not reported in our main paper; they are available in our repository). Together, our data strongly suggest that most characteristics were stereotyped and that gender stereotypes change over development. Even given our stringent criteria, 43 characteristics showed significant age by gender interactions for ratings of typicality, and 22 characteristics

showed significant age by gender interactions for ratings of desirability (We wish to note, of course, that "desirability" is a complex construct and certainly not a construct that is likely to be fully addressed via a single likert-scale question. We use this terminology in order to most closely approach the ways in which previous work (e.g., Prentice & Carranza, 2002) has discussed prescriptive and proscriptive gender stereotypes.). These data highlight the importance of taking a developmental approach to studying gender stereotypes. After all, our theories of the development of gender stereotypes will necessarily differ depending on whether a particular stereotype persists throughout the lifespan, emerges only in adulthood, peaks at the onset of puberty, or displays some other pattern.

Below, we discuss some of the more important developmental changes that we identified. Future research should further explore the nature of and mechanisms underscoring these changes. Additionally, we hope that other researchers will find our database immediately useful in informing the development of new research materials. For example, researchers interested in backlash targeting young adolescents will likely wish to manipulate the gender typicality of targets' traits, but could not previously be certain about which traits are actually viewed as gender stereotypic for this particular age group. Thus, our database now enables the development of evidence-based stimulus materials conveying the gender (counter)stereotypicality of targets across the developmental timespan.

As an important first step, we qualitatively and exploratorily classified the several developmental patterns that emerged in our data. These classifications were driven by the shape of the data and not by any theoretical expectations about *which* stereotypes might fall into each category. Because these clusters were generated exploratorily and subjectively, we encourage future researchers to use these primarily to motivate future confirmatory research and to generate testable theories. Further, a visual inspection of our data suggests the possibility that some of the developmental changes in gender stereotyping may be non-linear. The present study was not designed to differentiate possible developmental trajectories and did not sample ages densely enough to effectively do so (e.g., step-functions vs. logarithmic vs. quadratic vs. linear; see [29] for review). Thus, we encourage researchers to explore our dataset, and to conduct further research specifically aimed at detecting differences in the shape of change of gender stereotyping across the lifespan. In addition, we describe some of the qualitative patterns that emerged in our dataset below.

First, we note the presence of the *Gender Differences Emerge Late* category of stereotypes; these can be found in Fig 1 and towards the bottom of Tables 6 and 7. These are items for which we identified gender stereotypes in adulthood but found that these stereotypes were minimal or absent for younger targets (e.g., ambitious, submissive). These items are important because they shed light on the existing adult gender literature in that some of these items are critical to existing theories of gender development.

The fact that there are any characteristics for which the direction of a significant perceived gender gap switches throughout the lifespan (i.e., the *Gender Difference Switches* cluster) is a particularly novel and surprising revelation. Additionally, it is noteworthy that some gender stereotypes appear to be strongest in adolescence (i.e., the *Adolescent Gender Differences—Boost* and *Adolescent Gender Differences—Reduction* clusters) or in childhood (the *Childhood Gender Differences* cluster), or in only one age group (e.g., *Gender Differences Fluctuate)*. We have no theoretical account for these particular clusters at this moment and note that none of the clusters cleave neatly along existing theoretical lines.

Notably, we found 20 stereotypes that apply even to 1-year-old children. Of importance, from a purely developmental perspective, some of these characteristics could be difficult for children in this age group to display. For example, our data revealed that 1-year-old boys should be "athletic" and "love sports" and cannot "wear tutus" or "love pink." It is unclear how

a 1-year-old boy (who may not yet be walking and is unlikely to be talking) could adequately convey their athleticism and enthusiasm for sports, or their disdain for a color category they likely have no cognitive appreciation for and an article of clothing they are unlikely to have selected themselves. Similarly, 1-year-old girls should be "graceful" and "like princesses," and should not be "dirty" or "challenge authority." One might reasonably expect that infants and toddlers are too young to be constrained by these sorts of expectations, and that instead, adults would simply focus on whether very young children are healthy and meeting appropriate developmental milestones. Indeed, from this perspective, *it is noteworthy that any characteristics emerged as prescriptive and proscriptive for 1-year-olds*. Certainly, the current work makes the novel contribution of demonstrating that even infants appear to experience the effects of gender stereotyping.

Our results also clearly suggest that the stereotypes that individuals are faced with change over development. Of importance, existing theories of gender stereotypes were not created and thus, are unlikely to be able to account for these changes. For example, both backlash theory [30] and the Stereotype Content Model [23, 24] emphasize that adult men are expected to exhibit agentic, competence-related traits while women are expected to display communal, warmth-related traits. However, when examining the pre- and proscriptions, we uncovered that for young children it is not apparent that warmth and competence are the primary stereotypic dimensions relevant for classifying children. Instead, consistent with our prior work [13], young children's pre- and proscriptions appear to be much more linked to appearance (e.g., clothing choices) and overt, developmentally-relevant behaviors (e.g., play preferences). This suggests that it will be important to expand the existing literature to think more broadly about the nature, fluctuation, and impact of gender stereotypes throughout the lifespan. We hope that future researchers will utilize our developmental data to hone their theories of and predictions about the origins and time course of gender stereotypes.

While much of the adult literature has focused on stereotypes about women, our data show several ways in which boys and men may experience negative gender stereotyping. First, we found that the stereotypes that were considered *typical* of boys and men were also more often rated as unfavorable (see also [13]); this was true across the developmental timespan studied. Second, we found that across the timespan studied, there were more proscriptive stereotypes for boys/men than for girls/women. These results build upon a growing body of work demonstrating that gender stereotypes can have profound consequences for men as well as women (for a discussion, see [4]). For example, men appear to encounter backlash when they violate gender stereotypes by expressing interest in female gender-typed careers [5], behave modestly on a job interview [6], or disclose their emotions [31]. Further, recent work has shown that adults' reactions to 3-year-old boys who violate gender stereotypes may be particularly harsh relative to same-aged girls who violate gender stereotypes [13]. Taken together, these findings suggest that future work should continue to consider the ways in which gender stereotyping impacts perceptions of targets across the gender spectrum.

While the analyses reported here were pre-registered, we nevertheless consider them to be exploratory: we didn't have strong predictions about which stereotypes would persists across development (e.g., we found that boys/men are more rowdy and competitive, girls/women are more flatterable and caring), which peak in adolescence (e.g., girls cry more than boys; boys have a bigger appetite than girls), which would show stereotype vacillations across the timespan (e.g., boys/men are only sometimes more stingy than girls/women), and which would show no substantial gender stereotypes at all (e.g., neither gender is more bratty, stubborn, materialistic, rational, weak, or independent). While it may be tempting to believe that some of the developmental patterns that we demonstrate are the result of noise in the data, we believe it is unlikely that the patterns we see are false positives. First, our sample sizes are large:

each datapoint (e.g., the mean rating of typicality for *intelligent* for 3-year-olds boys) consists of 100 ratings. While it is possible for noise to be (erroneously) treated as signal, we believe that our high-powered design has likely revealed many provocative patterns of the development of gender stereotypes that were undetected in prior studies. Second, we pre-registered our data collection techniques and analyses and relied on measuring effect sizes (in addition to null hypothesis significance testing), reducing the likelihood that the patterns in our data emerged due to questionable research practices or because our design was overpowered. For these reasons, we hope future researchers will take seriously both the predicted and surprising developmental findings reported in our dataset.

We wish to note two major limitations of this study. First, we treated gender as a binary, when we know that gender is actually a continuum (see [32] for review). Tellingly, none of our participants noted any concerns about our binarization of gender. While we do not believe that a binarized view of gender is the right one, we do believe that the average adult in the United States assumes it to be, and our participants were familiar with and able to discuss gender in a binary way. To this point, we also note that we only sampled adults in the United States. We have no reason to believe that these stereotypes generalize to other cultural contexts, and indeed, this calls for additional cross-cultural research that can provide culturally-specific information about the content of descriptive, prescriptive, and proscriptive gender stereotypes across the lifespan.

In sum, we provide a novel and rich resource for future researchers cataloging the content of gender stereotypes across the early developmental timespan. The current results highlight the importance of expanding current theories of gender stereotyping to include developmental perspectives. Simply put, current theories of gender stereotyping may be specific to one point in development (i.e., adulthood). While this is useful for informing our understanding of the ways in which gender stereotypes about adults impact perceptions of adults, additional work is needed to shed light on the ways in which gender stereotypes shape and constrain social perceptions and experiences across the lifespan. Our analyses suggest several fruitful specific directions for new programs of research (e.g., focusing on the impacts of stereotyping on infants and their caregivers; emphasizing research on boys; using developmental trajectories to inform conceptual accounts of stereotyping), and we hope that researchers will use our dataset as a resource to inform both their theoretical and empirical future work.

## Author Contributions

**Conceptualization:** Jessica Sullivan, Corinne A. Moss-Racusin.

**Data curation:** Jessica Sullivan, Angela Ciociolo.

**Formal analysis:** Jessica Sullivan.

**Investigation:** Jessica Sullivan, Corinne A. Moss-Racusin.

**Methodology:** Jessica Sullivan, Corinne A. Moss-Racusin.

**Project administration:** Jessica Sullivan.

**Resources:** Corinne A. Moss-Racusin.

**Software:** Angela Ciociolo.

**Supervision:** Jessica Sullivan.

**Validation:** Jessica Sullivan.

**Visualization:** Jessica Sullivan, Angela Ciociolo.

**Writing – original draft:** Jessica Sullivan, Corinne A. Moss-Racusin.

**Writing – review & editing:** Jessica Sullivan, Angela Ciociolo, Corinne A. Moss-Racusin.

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
