## [Decision Letter · Decision Letter 0]

21 Jan 2021

PONE-D-20-25142

Establishing the Content of Gender Stereotypes Across the Lifespan

PLOS ONE

Dear Dr. Sullivan,

thank you for submitting your manuscript to PLOS ONE. After careful consideration by myself and two experts in the field, we feel that it has merit but does not fully meet PLOS ONE’s publication criteria as it currently stands. Therefore, we invite you to submit a revised version of the manuscript that addresses the points raised during the review process.

In your revision, please pay particular attention to the technical and statistical aspects of your work as mentioned in the Reviewwers' comments.  Also, statements of study primacy (such as "this is the first study..") have to be well-grounded.  So, please make sure to justify the original rationale of your work and discuss your work and findings in the context of existing literature such as Koenig (2018), making clear the novelty of your research. 

Please submit your revised manuscript not later than six months from this date as otherwise, any revision has to be considered a new submission. If you will need more time than this to complete your revisions, please reply to this message or contact the journal office at plosone@plos.org. Please include the following items when submitting your revised manuscript:

We look forward to receiving your revised manuscript.

Kind regards,

Sasha

Alexander N. 'Sasha' Sokolov, Ph.D.

Academic Editor

PLOS ONE

Journal Requirements:

Reviewers' comments:

Reviewer's Responses to Questions

**Comments to the Author**

1. Is the manuscript technically sound, and do the data support the conclusions?

Reviewer #1: Yes

Reviewer #2: Yes

2. Has the statistical analysis been performed appropriately and rigorously? 

Reviewer #1: I Don't Know

Reviewer #2: I Don't Know

3. Have the authors made all data underlying the findings in their manuscript fully available?

Reviewer #1: Yes

Reviewer #2: Yes

4. Is the manuscript presented in an intelligible fashion and written in standard English?

Reviewer #1: Yes

Reviewer #2: Yes

5. Review Comments to the Author

Reviewer #1: PONE-D-20-25142: Establishing the Content of Gender Stereotypes Across the Lifespan

I found this paper, which described gender stereotypes across childhood and adolescence, interesting and well-written. I would like to see the method and results of this work more directly compared to those of past studies that cataloged prescriptive and descriptive stereotypes – including Koenig (2018, who also assessed prescriptive and descriptive stereotypes across age categories, very similar to the current study) as well as Prentice (in terms of the results for adults). I believe this study adds information above these past studies, but it would be helpful if the method and results were compared to highlight how this study extends and builds on this past research. For example, I found it odd that the focus in the intro is on the stereotype content model for adults (pg. 11), which is about descriptive stereotypes, rather than this past work on prescriptive and descriptive stereotypes.

I applaud the researchers for preregistering the study, but I was unclear what exactly was preregistered. The first time preregistration is mentioned on page 13 says “preregistered yet exploratory findings” but it is unclear at this point in the paper what these exploratory analyses would be. It would be useful here to outline the three specific goals currently listed later on page 16, which would help guide the reader and understand what was preregistered.

How was the list of traits created? Some of the traits seems to apply better to children or to adults. What were participants to do when rating whether adults wear tutus (unlikely) or “gets pushed around by other kids” (did this say “other adults” when rating adults?), for example?

It might not be possible for space reasons, but it would be quite helpful if the graphs of the different “visualization pattern” of stereotypes over time were incorporated into the table listing the stereotypes with different patterns, as it is difficult to remember the various patterns. Perhaps one prototypical stereotype could be directly graphed as an example, rather than (what I assume are currently) hypothetical patterns. It was unclear how it was decided which characteristics fit each pattern – was this a visual inspection by the researchers (and if so, was this done individually and then the labels compared, as is done for qualitative codings?) or was there a quantitative analysis of similarity (or whether the effect was linear or quadratic, etc.)? Also, the characteristic “childlike” is listed twice in Table 6 under two different patterns.

I would also have liked more discussion of the findings in terms of, for example, the types/groups of traits that showed different patterns or results. Are there common themes (e.g., communion, agency, negative communion, negative agency, competence, physical characteristics, personality traits vs. behaviors) in the traits that show different results? This would make it a bit easier to digest the findings, as well as having the individual traits listed out for the specifics.

Reviewer #2: This manuscript reports a study of gender stereotypes across a variety of ages. This study appears to have been well-conducted overall. I have a few comments for the authors to consider as they move forward.

In several places, the manuscript refers to effects over "the lifespan" or "the developmental timespan," but 6 of the seven ages are children/adolescents, and the other one, 35, is used as a stand in for all adults. Truly examining this over the lifespan would sample more widely across adulthood and include older adult ages (e.g., 65, 85). Please reframe the description of these effects to more accurately reflect the represented ages.

Relatedly, for Figure 2, the caption says that the data suggest that children may be subjected to the most gender stereotypes, but again you have six ages for children and only one for all of adulthood! I think this is quite a stretch to say given the data.

It's unclear to me what "desirable" attributes are, as participants respond to these items. Are they saying they personally feel these attributes are desirable, or that they are seen in general as desirable? Do the authors have any validity evidence from this or other work using this wording that can tell us how best to understand how participants likely interpret "desirable" in this context? Literature on cultural vs. personal stereotypes would also be relevant to reference here.

The manuscript referred to preregistrations, which is great, but when I looked on OSF to see them I couldn't find them. This may be due to OSF's admittedly clunky navigation system (or my clunky inability to navigate it) but please do ensure that the preregistrations are made available.

The description of the analytical approach comes a little late - in the "Ratings of Typicality" section, but it looks like it was already employed to evaluate the "items that were not stereotyped" section. Please introduce the approach earlier to give context for how to interpret all reported results.

The Mturk sample is very large, which is great, but Mturk skews very White and likely other ways as well (e.g., politically liberal). Given that the stated aim is in part to create normed ratings of stereotypes, this should be explicitly acknowledged and discussed in the Discussion section.

There is some ambiguity about what analysis was performed, exactly (e.g., for "stereotypes that change over the developmental timespan," did the authors use regression to identify interactions, or only the descriptive method described in this paragraph?). Explicitly and clearly describing all analyses would be helpful. For this reason I chose "I don't know" about whether the analyses have been performed appropriately and rigorously, but with just a little more explanation, I would answer "Yes."

I particularly appreciate the focus on effect sizes. Could the authors explain their thinking behind choosing these particular cutoffs (e.g., why does a rating of 6 or higher count).

Social roles theory of gender stereotypes seems relevant here, especially with respect to its focus on agency and communion, which are close to warmth and competence. I would encourage the authors to reference this work if/as relevant.

The approach taken here (using established traits from literature on gender stereotypes) makes sense. For their future work, I would encourage the authors to consider whether a bottom-up approach could complement this work, as that could allow them to see if there are unique gender stereotypes that arise at different ages that are not well-captured by existing gender stereotypes (which could reflect only certain ages).

6. PLOS authors have the option to publish the peer review history of their article (what does this mean?). If published, this will include your full peer review and any attached files.

Reviewer #1: No

Reviewer #2: No

---

## [Author Response · Author response to Decision Letter 0]

12 May 2021

Detailed responses to reviewers are included in our cover letter.

---

## [Decision Letter · Decision Letter 1]

13 Jul 2021

PONE-D-20-25142R1

Establishing the Content of Gender Stereotypes Across Development

PLOS ONE

Dear Dr. Sullivan,

thank you for submitting your revised manuscript to PLOS ONE.  After careful consideration, we feel that it has merit but as it currently stands, still has to be improved to fully meet PLOS ONE’s publication criteria.  I appreciate very much the efforts of two experts in the filed who have provided their detailed and valuable feedback on your revision, and both been positive about your work.  Therefore, we invite you to submit a revised version of the manuscript that addresses the points raised during the review process, paying particular attention to the following points below: 

1) As you will see from the comments, both Reviewwers have had difficulty in accessing documents mentioned as pre-registered.  Please provide a correct link to the files.

2) Please make sure to address the technical and statistical issues raised by both Reviewers such as multiplicity corrections and statistical trend analyses.  In addition, please (a) state if data sets were tested for normality and which (parametric, nonparametric) statistical inference was used for respective sets, (b) which statistic (and directional or not) was used for each p-value, and (c) add variability measures where appropriate.

Also please respond thoroughly to the other concerns expressed by the Reviewers. 

Please submit your revised manuscript within six moinths from this date as after that any revision has to be considered a new submission.  If you will need more time than this to complete your revisions, please reply to this message or contact the journal office at plosone@plos.org. Please include the following items when submitting your revised manuscript:

We look forward to receiving your revised manuscript.

Thank you for choosing PLOS ONE for communicating your research.

Kind regards,

Sasha

Alexander N. 'Sasha' Sokolov, Ph.D.

Academic Editor

PLOS ONE

Reviewers' comments:

Reviewer's Responses to Questions

**Comments to the Author**

1. If the authors have adequately addressed your comments raised in a previous round of review and you feel that this manuscript is now acceptable for publication, you may indicate that here to bypass the “Comments to the Author” section, enter your conflict of interest statement in the “Confidential to Editor” section, and submit your "Accept" recommendation.

Reviewer #1: All comments have been addressed

Reviewer #2: (No Response)

2. Is the manuscript technically sound, and do the data support the conclusions?

Reviewer #1: Yes

Reviewer #2: Yes

3. Has the statistical analysis been performed appropriately and rigorously? 

Reviewer #1: Yes

Reviewer #2: Yes

4. Have the authors made all data underlying the findings in their manuscript fully available?

Reviewer #1: Yes

Reviewer #2: Yes

5. Is the manuscript presented in an intelligible fashion and written in standard English?

Reviewer #1: Yes

Reviewer #2: Yes

6. Review Comments to the Author

Reviewer #1: The authors have revised the manuscript to address the reviewer’s concerns, making the paper even stronger and clearer. The data set is large, but the authors do a good job of aligning the analysis of the data with their goals and distilling the main points of the findings. I appreciate the thoroughness of their response to the previous reviews. I have only a few very minor suggestions for the authors to take into consideration as they move forward with editing, to make the paper as clear as possible.

Pg. 5-6: “because it elicited ratings only for developmental categories (e.g., “adolescents (ages 12-18)”),” This sentence is not complete.

Pg. 6: “in order for our general theories of gender stereotyping (e.g., Social Role Theory (e.g., Eagly & Wood, 2012; 2016) and the Stereotype Content Model; Fiske et al., 2002; Fiske et al., 1999) to be useful, they must fit the data not only for adults but also for children.” I’m not sure the authors want to imply these theories are not useful, as they were designed for stereotypes of adults and have created meaningful bodies of research. Certainly, it's possible they may not apply well to stereotypes of children (or it could be that the social roles of children do impact their gender stereotypes, as would be suggested by social role theory, which doesn't mean the stereotypes have to mirror adult stereotypes). So this sentence could be reworded to suggest these theories may need to be adapted or new theories created to address gender stereotypes across development, without throwing out these theories entirely. Otherwise, this seems like a strong point – to say these theories would not be useful if gender stereotypes differ across ages – that would need more elaboration and discussion that this one sentence.

Pg. 12: The list of traits that are not prescriptive for gender includes three items that reference "kids" – gets pushed around by other kids, pulls other kids’ hair, thinks it’s funny when other kids are crying. It is unclear whether these items were used verbatim for all groups (even adolescents and adults) or whether the item changed to reflect the age group that was being rated (e.g., “gets pushed around by other adults”).

Given the multiple tests of comparison, it would be useful to include a statement at the beginning of the results that indicates the criterion for labeling something as a significant main effect or interaction (was it p < .05?). This may be stated in the preregistration, but the links provided for the preregistration took me only to the data and I did not see a preregistration document there (although perhaps I did not know where to look - but I suggest checking the link to make sure it leads to the preregistration documents). I know it is the effect size that qualifies a characteristics as a stereotype or not, but it is discussed how many traits showed significant effects, so knowing this cutoff is still relevant.

Pg. 17: The authors reference “In our previous work, we demonstrated that..” but do not give a citation (likely for blinding for review). I would suggest now adding in a citation here so that readers know what work is being referenced.

Table 6: Perhaps in the “direction switches” category it could be indicated which gender was higher at the youngest ages and then which was high later (the “switch”), instead of indicating “N/A.”

Table 8 has blanks when the stereotype was not relevant to that age group, but Table 9 uses dashes (and one blank). Do these dashes also represent ages where the stereotype was not relevant? I would suggest using blanks in both tables as that seems the most intuitive (and I believe APA style suggests dashes when the data are not collected/irrelevant, which is not the case here).

Reviewer #2: I was a reviewer on the prior submission. I really like the aim of this project – to descriptively catalogue gender stereotypes at a range of ages. The current manuscript is stronger and has overall addressed my comments well. I have a few comments that I hope will help to strengthen the paper further.

In response to my original comment about the pre-registration (#17), I appreciate the clarification, but I still can’t find the files. When I go to the OSF page, at first the only options available to me are “Files” “Wiki” and “Analytics”. Then when clicking “Analytics”, “Registrations” appears. But then when I click on “Registrations”, I get a message “there was an error loading this list”. So, I still have not been able to review the registrations.

I really appreciate the authors moving away from phrases like “lifespan” and being clearer in identifying the ages examined in this work. However, the phrase “across the developmental timespan” (e.g., as used in the abstract and the introduction) still implies that the developmental timespan encompasses only childhood / up to middle adulthood, when in fact development continues throughout the lifespan. Perhaps a phrase like “early lifespan” or “early developmental timespan” or “from childhood through early adulthood” would be more accurate?

In the method: “For example, participants in one condition saw: “Today you will be

answering questions about how [common or typical/desirable] you think particular traits are among 1-year-old boys”.”

Should either “common or typical” or “desirable” be listed, as it’s an example, or did participants within this condition actually rate on both commonness and desirability (which is what this wording could suggest)?

I really like the visualization for the interactions. But, only a linear term of age is used, which could potentially omit some characteristics that would have quadratic/cubic effects for one gender but not another. If there are nonlinear patterns of interaction e.g., if girls tend to increase and then decrease on a trait, whereas for boys it remains flat, that won’t necessarily appear in the interactions as specified if the overall (linear) change for girls is steady. It looks from the visualizations that for many of those characteristics that showed a difference in the linear slope (i.e., a significant interaction), there was likely some difference in quadratic/cubic change, too – e.g., childhood gender differences, adolescent gender differences – boost, and fluctuating gender differences. These traits just happen to have been identified by the linear interactions because there is also an overall difference in slope between genders, and not because the models run could identify these kinds of patterns. This suggests that there may be characteristics for which males and females differ in their quadratic/cubic patterns but not in their overall linear term. Could the authors re-rerun the regressions and see if including a quadratic or cubic term for age would suggest additional characteristics to explore? This can be done by adding both age(centered)^2 as a linear term and as interacting with gender, and age(centered)^3 as a linear term and as interacting with gender. This could potentially identify additional characteristics that fit the current set of visualization patterns, and/or new ones.

Regarding: “Critically, these findings are unlikely to be due to biases in the characteristics tested in our study: if they were, we would not expect the symmetrical rate of descriptive stereotypes.” (p. 28).

I don’t quite follow this, can the authors provide a little more explanation of the logic here?

7. PLOS authors have the option to publish the peer review history of their article (what does this mean?). If published, this will include your full peer review and any attached files.

Reviewer #1: No

Reviewer #2: No

---

## [Author Response · Author response to Decision Letter 1]

23 Sep 2021

Reviewer #1

Reviewer #1: The authors have revised the manuscript to address the reviewer’s concerns, making the paper even stronger and clearer. The data set is large, but the authors do a good job of aligning the analysis of the data with their goals and distilling the main points of the findings. I appreciate the thoroughness of their response to the previous reviews. I have only a few very minor suggestions for the authors to take into consideration as they move forward with editing, to make the paper as clear as possible.

We thank the reviewer for their kind words, and appreciate the remaining suggestions, which we have responded to below.

Pg. 5-6: “because it elicited ratings only for developmental categories (e.g., “adolescents (ages 12-18)”),” This sentence is not complete.

The sentence now reads “This study provides an exciting and promising window into understanding how gender stereotypes differ across the lifespan; however, because it elicited ratings only for developmental categories (e.g., “adolescents (ages 12-18)”), it does not allow us to draw conclusions about developmental trajectories.”

Pg. 6: “in order for our general theories of gender stereotyping (e.g., Social Role Theory (e.g., Eagly & Wood, 2012; 2016) and the Stereotype Content Model; Fiske et al., 2002; Fiske et al., 1999) to be useful, they must fit the data not only for adults but also for children.” I’m not sure the authors want to imply these theories are not useful, as they were designed for stereotypes of adults and have created meaningful bodies of research. Certainly, it's possible they may not apply well to stereotypes of children (or it could be that the social roles of children do impact their gender stereotypes, as would be suggested by social role theory, which doesn't mean the stereotypes have to mirror adult stereotypes). So this sentence could be reworded to suggest these theories may need to be adapted or new theories created to address gender stereotypes across development, without throwing out these theories entirely. Otherwise, this seems like a strong point – to say these theories would not be useful if gender stereotypes differ across ages – that would need more elaboration and discussion that this one sentence.

Thank you for pointing out this unfortunate wording. We do not wish to imply that the SCM isn’t useful. We now say: “First, in order for our general theories of gender stereotyping (e.g., Social Role Theory (e.g., Eagly & Wood, 2012; 2016) and the Stereotype Content Model; Fiske et al., 2002; Fiske et al., 1999) to be useful for understanding and predicting children’s learning about gender stereotypes, they must fit the data not only for adults but also for children”

Pg. 12: The list of traits that are not prescriptive for gender includes three items that reference "kids" – gets pushed around by other kids, pulls other kids’ hair, thinks it’s funny when other kids are crying. It is unclear whether these items were used verbatim for all groups (even adolescents and adults) or whether the item changed to reflect the age group that was being rated (e.g., “gets pushed around by other adults”).

Thank you for pointing out this ambiguity. These items were used verbatim.

Given the multiple tests of comparison, it would be useful to include a statement at the beginning of the results that indicates the criterion for labeling something as a significant main effect or interaction (was it p < .05?). This may be stated in the preregistration, but the links provided for the preregistration took me only to the data and I did not see a preregistration document there (although perhaps I did not know where to look - but I suggest checking the link to make sure it leads to the preregistration documents). I know it is the effect size that qualifies a characteristics as a stereotype or not, but it is discussed how many traits showed significant effects, so knowing this cutoff is still relevant.

We apologize again for the error in the preregistration link. Our alpha was .05, and all tests were two-tailed. Our effect-size cutoffs were also pre-registered.

Pg. 17: The authors reference “In our previous work, we demonstrated that..” but do not give a citation (likely for blinding for review). I would suggest now adding in a citation here so that readers know what work is being referenced.

Thank you. We have now added our citation.

Table 6: Perhaps in the “direction switches” category it could be indicated which gender was higher at the youngest ages and then which was high later (the “switch”), instead of indicating “N/A.”

When we attempted to implement this suggestion, we found that it looked quite clunky (e.g., “higher for girls at ages 1 and 4, and higher for boys for age 7+”). In addition, the analyses required to determine the precise nature of the “switch” are different from the analyses used in the rest of the table (the former involves pairwise comparisons within age, while the latter involves regression models over the entire dataset); we were concerned that reporting particular age information in this table might confuse readers.

However, we agree with the reviewer that “N/A” is not the ideal notation for this table. We now say “direction switches” in this column. Readers who are interested in understanding the precise nature of the switch can access our data and supplemental materials.

Table 8 has blanks when the stereotype was not relevant to that age group, but Table 9 uses dashes (and one blank). Do these dashes also represent ages where the stereotype was not relevant? I would suggest using blanks in both tables as that seems the most intuitive (and I believe APA style suggests dashes when the data are not collected/irrelevant, which is not the case here).

We have now made our formatting consistent, by changing all dashes to blanks.

Reviewer #2: I was a reviewer on the prior submission. I really like the aim of this project – to descriptively catalogue gender stereotypes at a range of ages. The current manuscript is stronger and has overall addressed my comments well. I have a few comments that I hope will help to strengthen the paper further.

Thank you! We really appreciate the time you’ve put into improving the manuscript.

In response to my original comment about the pre-registration (#17), I appreciate the clarification, but I still can’t find the files. When I go to the OSF page, at first the only options available to me are “Files” “Wiki” and “Analytics”. Then when clicking “Analytics”, “Registrations” appears. But then when I click on “Registrations”, I get a message “there was an error loading this list”. So, I still have not been able to review the registrations.

We are really sorry -- we (accidentally!) had two OSF pages for the same project with very similar titles; one had the data and the other had the preregistration. The correct preregistration link is now throughout the article, and we have moved all datasets etc… from the other OSF page to the correct one. As noted above, the correct link to the pre-registration is: https://osf.io/7ahks and the correct link the OSF page containing that pre-registration is: https://osf.io/9pgkd/

I really appreciate the authors moving away from phrases like “lifespan” and being clearer in identifying the ages examined in this work. However, the phrase “across the developmental timespan” (e.g., as used in the abstract and the introduction) still implies that the developmental timespan encompasses only childhood / up to middle adulthood, when in fact development continues throughout the lifespan. Perhaps a phrase like “early lifespan” or “early developmental timespan” or “from childhood through early adulthood” would be more accurate?

Thank you. We have changed it to “early developmental timespan” in most cases, and to the “developmental timepsan studied” when we are referring to the ages tested in our study.

In the method: “For example, participants in one condition saw: “Today you will be

answering questions about how [common or typical/desirable] you think particular traits are among 1-year-old boys”.”Should either “common or typical” or “desirable” be listed, as it’s an example, or did participants within this condition actually rate on both commonness and desirability (which is what this wording could suggest)?

It now says “...questions about how [“common or typical”][“desirable”] you think particular traits are among…”

I really like the visualization for the interactions. But, only a linear term of age is used, which could potentially omit some characteristics that would have quadratic/cubic effects for one gender but not another. If there are nonlinear patterns of interaction e.g., if girls tend to increase and then decrease on a trait, whereas for boys it remains flat, that won’t necessarily appear in the interactions as specified if the overall (linear) change for girls is steady. It looks from the visualizations that for many of those characteristics that showed a difference in the linear slope (i.e., a significant interaction), there was likely some difference in quadratic/cubic change, too – e.g., childhood gender differences, adolescent gender differences – boost, and fluctuating gender differences. These traits just happen to have been identified by the linear interactions because there is also an overall difference in slope between genders, and not because the models run could identify these kinds of patterns. This suggests that there may be characteristics for which males and females differ in their quadratic/cubic patterns but not in their overall linear term. Could the authors re-rerun the regressions and see if including a quadratic or cubic term for age would suggest additional characteristics to explore? This can be done by adding both age(centered)^2 as a linear term and as interacting with gender, and age(centered)^3 as a linear term and as interacting with gender. This could potentially identify additional characteristics that fit the current set of visualization patterns, and/or new ones.

We agree that this is a fascinating possibility, and one that is ripe for examination in future research. However, comparing models in these suggested ways would add a substantial amount of complexity and information to the paper, which is already dense. Adding to the complexity is that these analyses would not only be exploratory, but also atheoretical: while we acknowledge that there are some visible non-linearities in our data, we know of no theories that predict a particular cubic or quadratic effect of age -- or, critically, an interaction of target gender with a cubic or quadratic age function. This makes it challenging to explore this question in a targeted way -- when conducting exploratory analyses that deviate from the plan, it is best practice to do so when theoretically motivated.

More generally, our ability to make fine-grained determinations about the shape of developmental change hinges on the density of our age sampling; while our sampling of different ages across childhood allows us to make general statements about the overall trajectory, without fine-grained sampling of ages (i.e., at least of every year), any descriptions of the shape of developmental change will be impacted substantially by our sampling frequency (for an excellent description and demonstration of this in the domain of motor development, see Adolph et al., 2008). 

Our goal is to provide some preliminary observations about trends in these data, and then make the dataset available for other researchers to examine in numerous different ways. As such, we have added the sentence to the General Discussion on pp. 31-32. :

 “Further, a visual inspection of our data suggests the possibility that some of the developmental changes in gender stereotyping may be non-linear. The present study was not designed to differentiate possible developmental trajectories and did not sample ages densely enough to effectively do so (e.g., step-functions vs. logarithmic vs. quadratic vs. linear; see Adolph, 2008 for review). Thus, we encourage researchers to explore our dataset, and to conduct further research specifically aimed at detecting differences in the shape of change of gender stereotyping across the lifespan. In addition, we describe some of the qualitative patterns that emerged in our dataset below”

Regarding: “Critically, these findings are unlikely to be due to biases in the characteristics tested in our study: if they were, we would not expect the symmetrical rate of descriptive stereotypes.” (p. 28). I don’t quite follow this, can the authors provide a little more explanation of the logic here?

Upon revisiting that paragraph, we also are unsure what we intended with this sentence -- we have removed it from the manuscript.

---

## [Decision Letter · Decision Letter 2]

17 Jan 2022

Establishing the Content of Gender Stereotypes Across Development

PONE-D-20-25142R2

Dear Dr. Sullivan,

thank you for your revised manuscript above.  We’re pleased to inform you that your manuscript has been judged scientifically suitable for publication and will be formally accepted for publication once it meets all outstanding technical requirements.

Thank you for submitting your research to PLOS ONE.

Kind regards and stay safe and healthy in 2022,

Sasha

Alexander N. 'Sasha' Sokolov, Ph.D.

Academic Editor

PLOS ONE

Additional Editor Comments (optional):

Reviewers' comments:

Reviewer's Responses to Questions

**Comments to the Author**

1. If the authors have adequately addressed your comments raised in a previous round of review and you feel that this manuscript is now acceptable for publication, you may indicate that here to bypass the “Comments to the Author” section, enter your conflict of interest statement in the “Confidential to Editor” section, and submit your "Accept" recommendation.

Reviewer #1: All comments have been addressed

Reviewer #2: All comments have been addressed

2. Is the manuscript technically sound, and do the data support the conclusions?

Reviewer #1: Yes

Reviewer #2: Yes

3. Has the statistical analysis been performed appropriately and rigorously? 

Reviewer #1: Yes

Reviewer #2: Yes

4. Have the authors made all data underlying the findings in their manuscript fully available?

Reviewer #1: Yes

Reviewer #2: Yes

5. Is the manuscript presented in an intelligible fashion and written in standard English?

Reviewer #1: Yes

Reviewer #2: Yes

6. Review Comments to the Author

Reviewer #1: (No Response)

Reviewer #2: (No Response)

7. PLOS authors have the option to publish the peer review history of their article (what does this mean?). If published, this will include your full peer review and any attached files.

Reviewer #1: No

Reviewer #2: **Yes: **Rebecca Neel

---

## [Editor Report · Acceptance letter]

15 Feb 2022

PONE-D-20-25142R2 

Establishing the Content of Gender Stereotypes Across Development 

Dear Dr. Sullivan:

I'm pleased to inform you that your manuscript has been deemed suitable for publication in PLOS ONE. Congratulations! Your manuscript is now with our production department. 

Kind regards, 

on behalf of

Dr. Alexander N. Sokolov 

Academic Editor

PLOS ONE